# Attachment Orientation and Preferences for Partners’ Emotional Responses in Stressful and Positive Situations

**DOI:** 10.3390/bs14010077

**Published:** 2024-01-22

**Authors:** Brian N. Chin, Lauryn Kim, Shelby M. Parsons, Brooke C. Feeney

**Affiliations:** 1Department of Psychology, Trinity College, Hartford, CT 06106, USA; 2Department of Psychology, Carnegie Mellon University, Pittsburgh, PA 15213, USAbfeeney@andrew.cmu.edu (B.C.F.)

**Keywords:** close relationships, relational regulation, emotion regulation, social support, affect

## Abstract

Attachment theory proposes that close relationships help us to regulate our emotions in stressful and positive situations. However, no previous studies have examined preferences for a partner’s emotional response to one’s own stressful and positive situations or tested whether these preferences differ based on attachment orientation. This study examines the association of attachment orientation and preferences for partners’ emotional responses relative to one’s own emotional responses in stressful and positive contexts among 425 United States adults who were currently in a committed relationship of ≥6 months. Data were collected in 2020. Overall, participants preferred their partners to feel and express less distress, less worry, more calm, and more hope than themselves during stressful situations and for their partners to feel and express more excitement, pride, and hope than themselves during positive situations. Higher attachment anxiety predicted preferences for partners to feel and express more distress/worry in stressful situations, whereas higher attachment avoidance predicted preferences for partners to feel and express less hope in stressful situations. Statistical interactions of attachment anxiety × attachment avoidance indicated that the combination of low attachment anxiety and high attachment avoidance (dismissing avoidance) was associated with preferences for partners to feel and express less positive emotions in positive situations, whereas the combination of high attachment anxiety and high attachment avoidance (fearful avoidance) was associated with preferences for partners to feel and express more negative emotions in stressful situations and less positive emotions in positive situations. This investigation provides novel evidence for links between attachment orientation and preferences for partners’ emotional responses in two theoretically important contexts, which has implications for the nature and function of emotion regulation in close relationships. Future research is needed to determine the generalizability of these findings to more collectivist cultural contexts.

## 1. Introduction

We often turn to our close relationships to cope with stressful events or to celebrate good news. Sometimes, relationship partners provide emotional support by expressing the same emotions we are experiencing. Other times, partners provide support by calming us down from intense negative emotions or by helping us enjoy and amplify positive emotions. Although there have been numerous studies demonstrating the benefits of receiving support from one’s partner in both stressful and positive contexts [1], there has been no research focused on preferences for partners’ emotional responses to one’s own stressful and positive situations. Understanding preferences for a partner’s emotional responses relative to one’s own could provide insight into the ways that individuals regulate (e.g., amplify, maintain, dampen) their emotions with close partners in stressful and positive situations [2]. To address this gap in the literature, we examined attachment-related individual differences in preferences for partners’ emotional responses to stressful and positive situations relative to one’s own emotional responses.

### Attachment Orientation and Emotions

We used attachment theory as a theoretical framework for investigating individual differences in preferences for partners’ emotional responses to one’s own stressful and positive situations, relative to one’s own emotional responses to these situations. According to attachment theory, humans possess an innate propensity to form strong and enduring emotional bonds with close others (i.e., attachment figures) [3]. Experiences with one’s attachment figures (i.e., the extent to which needs for security are met), particularly in contexts involving stressors and positive contexts involving novel exploration, underlie the development of individual differences in attachment orientation [3,4]. Individual differences in attachment orientation have been identified across the lifespan and have been shown to influence the way one experiences and regulates one’s emotions in the presence of close others [2,5]. In the current investigation, we extended this literature by conducting a novel test of whether individual differences in attachment orientation are associated with preferences for partners’ emotional responses in two contexts through which adult attachment relationships facilitate optimal well-being—stressful situations and positive situations involving exploration behavior (e.g., goals, opportunities, accomplishments) [1].

Individual differences in adult attachment orientation are measured in two dimensions—attachment anxiety and attachment avoidance. Attachment anxiety is associated with a preoccupation with close others, hypervigilance to signs of rejection or abandonment by close others, and a worry that close others will be unavailable or unresponsive during times of need. Anxiously attached individuals frequently engage in emotional hyperactivation strategies that are intended to elicit reassurance or affection from close others, such as by heightening their expression of negative emotions during interactions with their partners [6,7]. Attachment avoidance is associated with compulsive self-reliance, mistrust of others, and the belief that close others will be unavailable and unresponsive in times of need. Avoidantly attached individuals prefer to engage in emotional suppression strategies that allow them to avoid relying on others and keep their attachment needs deactivated, such as minimizing their expression of negative emotions during interactions with their partners [2,8].

There is considerable evidence for attachment-related individual differences in one’s own emotional responses to stressful events. Past research has generally demonstrated that anxious attachment is associated with hyperactivation of negative emotions in stressful contexts, whereas avoidant attachment is associated with deactivation of negative emotions in stressful contexts [2]. For example, studies of emotional responses to recalled negative memories demonstrate that anxiously attached individuals more readily recall negative memories and report experiencing more intense negative emotions during recall, whereas avoidantly attached individuals less readily recall negative memories and report experiencing less intense negative emotions during recall [9]. Moreover, there is evidence that both trait attachment anxiety and experimentally induced attachment anxiety are associated with a greater propensity for experiencing false memories of relational stimuli [10,11]. Other studies have found attachment-related differences in how individuals attend to negative stimuli, with anxious individuals preferentially attending to negative stimuli [12,13] and avoidant individuals tending to suppress attention to negative stimuli [14]. Earlier investigations have also found that attachment anxiety and attachment avoidance are both associated with lower optimism and more catastrophic appraisals of potential threats [15]. Especially relevant to this investigation, Simpson et al. [16] demonstrated that securely attached individuals were most soothed by their partner’s emotional caregiving behaviors during relationship conflict, whereas avoidantly attached individuals responded more favorably to instrumental caregiving behaviors.

There is also evidence for attachment-related individual differences in one’s own emotional responses to positive events. Studies of attachment-related differences in emotional response to positive events have generally demonstrated that both anxious attachment and avoidant attachment interfere with the experience and expression of positive emotions [17]. For example, laboratory studies using emotion-induction procedures and facial coding have found that avoidantly attached individuals show less positive facial affect when viewing positive stimuli [18,19], whereas anxiously attached individuals show more negative facial affect when viewing positive stimuli (i.e., happy faces) [20]. Similarly, a daily diary study indicated that anxious attachment and avoidant attachment are both associated with blunted positive emotional responses to positive daily events [21].

## 2. Current Research

We aimed to build on past research examining attachment and emotional responses by investigating whether attachment orientation predicts individuals’ preferences for partners’ emotional responses, relative to their own emotional responses, in stressful situations (Aim 1) and positive situations (Aim 2). We conducted an online survey study of individuals who were currently in a committed romantic relationship that had been ongoing for at least six months. Participants reflected on times when they discussed stressful situations and positive situations with their romantic partners and indicated their preferences for their partners’ emotional responses (relative to their own responses) in those situations. We assessed participants’ preferences about two aspects of partners’ emotional responses to each type of situation—their partner’s emotional experience (felt emotion) and their partner’s emotional expression (expressed emotion). We assessed preferences for specific emotions that were theoretically relevant to stressful situations (distress, worry, calm, hope) and positive situations (excitement, pride, hope, calm). We tested the following specific hypotheses based on the postulates of attachment theory and past research on adult attachment orientation and emotions. 

First, we predicted that attachment anxiety would be associated with preferences for partners to respond to one’s stressful situations by feeling and expressing more distress, more worry, less calm, and less hope than themselves (Hypothesis 1a). We also predicted that attachment avoidance would be associated with preferences for partners to respond to one’s stressful situations by feeling and expressing less distress, less worry, less calm, and less hope than themselves (Hypothesis 1b). We hypothesized that these preferences would help anxious individuals, who tend to hyperactivate their negative emotions in stressful situations, and avoidant individuals, who tend to deactivate their positive and negative emotions in stressful contexts, to accomplish their emotion regulation goals in stressful situations.

Second, we predicted that attachment anxiety would be associated with preferences for partners to respond to one’s positive situations by feeling and expressing more excitement, more pride, more hope, and less calm than themselves (Hypothesis 2a). We also predicted that attachment avoidance would be associated with preferences for partners to respond to one’s positive situations by feeling and expressing less excitement, less pride, less hope, and more calm than themselves (Hypothesis 2b). We hypothesized that these preferences would help anxious individuals, who desire their partner’s approval and validation, and avoidant individuals, who prefer to keep their attachment needs deactivated, to accomplish their emotion regulation goals in positive situations.

## 3. Method

### 3.1. Participants and Procedures

Participants were 425 adults recruited through Amazon Mechanical Turk (*n* = 253) and Volunteer Science (*n* = 172) for a study of romantic relationships. Participants’ ages ranged from 18 to 88 years (*M* = 41.3, *SD* = 12.9). In total, 51.5% identified as female, 47.5% as male, and 0.9% as genderqueer or nonbinary. In total, 56.7% of participants identified as White, 32.0% as Asian, 3.5% as Black, 3.5% as multiracial, 2.8% as Hispanic, 0.5% as Native/Indigenous, and 0.7% as another race; 1 participant preferred not to disclose their race. In total, 79.8% of participants were cohabiting with their romantic partner. Sample size was determined based on the desire to maximize power by collecting data from as many participants as possible; data collection was halted when funding was no longer available. Data were collected between March and December 2020. Inclusion criteria were being ≥18 years old, fluent in English, and in a committed romantic relationship of at least six months. All study procedures were approved by a university institutional review board and complied with APA ethical standards for the treatment of human participants.

After providing informed consent, participants completed questionnaires via Qualtrics assessing their demographic information, attachment orientation, and preferences for being supported by romantic partners in stressful and positive situations. Participants were compensated USD 3 for completing the study via Amazon Mechanical Turk; participants did not receive compensation for completing the study via Volunteer Science.

We conducted a post hoc power analysis using G*Power version 3.1.9.7 [22] to compute achieved power for a linear regression with two tested predictors, a significance criterion of α = 0.05, and sample size = 425. Achieved power was 1.00 to detect a medium effect (f^2^ = 0.15) and 0.74 to detect a small effect (f^2^ = 0.02).

### 3.2. Measures

#### 3.2.1. Attachment Orientation

We assessed general attachment orientation using a 14-item version of the Experiences in Close Relationships (ECR) scale [23]. Participants rated the extent to which they agreed with statements assessing attachment anxiety (seven items, e.g., “I worry about being abandoned”) and attachment avoidance (seven items, e.g., “I try avoiding getting too close to people”) in their global attachment relationships (1 = strongly disagree, 7 = strongly agree). Attachment anxiety (α = 0.92) and attachment avoidance (α = 0.88) composites were computed by averaging the scores of items for each subscale.

#### 3.2.2. Preferences for Partners’ Emotional Responses to Stressful Situations

Participants were prompted to reflect on times when they discussed their own stressful situations with their romantic partner and rate how they wanted their partner to respond in these situations:


*“Please take a moment to think about times in your life when you have been distressed by significant, major stressors. For example, you might have been worried about a serious health problem that you or a family member was having, you might have performed poorly at something that was very important to you, you might have been treated badly at work, you may be having a hard time finding a job or getting accepted into an academic program, etc. These should be significant personal stressors that were NOT caused by your partner. When discussing these stressors with your partner, to what extent do you typically prefer that your partner respond in the following ways? Some questions below ask about two different aspects of your emotional life. One is your emotional experience, or what you feel inside. The other is your emotional expression, or how you communicate your emotions to others in the way you talk, gesture, or behave.”*


Participants rated four items assessing the extent to which they preferred their partner to feel distressed, calm, hopeful, and worried about the stressor relative to their own feelings (−3 = much less than me, 0 = as much as me, 3 = much more than me). Participants also rated four items assessing the extent to which they preferred their partner to express distress, calmness, hopefulness, and worry about the stressor relative to their own expressions of each emotion (−3 = much less than me, 0 = as much as me, 3 = much more than me). 

#### 3.2.3. Preferences for Partners’ Emotional Responses to Positive Situations

Participants were prompted to reflect on times when they discussed their own positive situations or events with their romantic partner and rate how they wanted their partner to respond in these situations:


*“Please take a moment to think about times in your life when you have felt excited about something positive. For example, you may have received good news about getting a new job or receiving an award or a promotion, or you may have an exciting new hobby or goal or opportunity to pursue. When discussing these positive experiences with your partner, to what extent do you typically prefer that your partner respond in the following ways? Some questions below ask about two different aspects of your emotional life. One is your emotional experience, or what you feel inside. The other is your emotional expression, or how you communicate your emotions to others in the way you talk, gesture, or behave.”*


Participants rated four items assessing the extent to which they preferred their partner to feel excitement, calm, pride, and hope about the positive situation relative to their own feelings (−3 = much less than me, 0 = as much as me, 3 = much more than me). Participants also rated four items assessing the extent to which they preferred their partner to express excitement, calm, pride, and hope about the positive situation relative to their own expressions of each emotion (−3 = much less than me, 0 = as much as me, 3 = much more than me). 

## 4. Data Analysis

Our descriptive analyses included bivariate correlations between predictor and outcome variables and paired samples t-tests evaluating whether participants differed in their preferences for partners to feel vs. express distress, worry, calm, and hope in stressful situations or in their preferences for partners to feel vs. express excitement, pride, calm, and hope in positive situations.

To address our first aim, we used linear regression models to test associations of attachment orientation (attachment anxiety, attachment avoidance, and the interaction of attachment anxiety × attachment avoidance) and preferences for partners’ emotional responses to stressful situations (feeling and expressing distress, worry, calm, and hope) relative to their own feelings and expressions.

To address our second aim, we used linear regression models to test associations of attachment orientation and preferences for partners’ emotional responses to positive situations (feeling and expressing excitement, calm, pride, and hope) relative to their own feelings and expressions.

To interpret statistically significant interactions of attachment anxiety × attachment avoidance, we conducted planned follow-up analyses of simple slopes of attachment anxiety at one standard deviation above and below the mean value of attachment avoidance.

We conducted these analyses with and without controlling for participants’ gender. Because all associations were unaffected by the inclusion of gender as a covariate, we only report the results of the more parsimonious models that did not control for gender.

### Transparency and Openness

We report how our sample size was determined and all data exclusions, manipulations, and measures in this study. Data were analyzed using SPSS, version 28.0.1.1, and the PROCESS macro, version 4.2, model 1 [24]. This study was not pre-registered. Data and analysis code are available at https://osf.io/nkmbw/?view_only=bb9aa20a40fc4e26a89866ed7b264efc (accessed on 20 November 2023). 

## 5. Results

### Descriptive Analyses

Descriptive statistics for preferred partner emotions in stressful situations and positive situations are shown in Table 1. Bivariate correlations of primary outcome variables are shown in Table 2.

As shown in Table 1, participants generally preferred their partners to feel and express less distress, less worry, more calm, and more hope relative to themselves during stressful situations and for their partners to feel and express more excitement, pride, and hope relative to themselves during positive situations. Participants also generally preferred their partners to feel more calm, but not express more calm, relative to themselves during positive situations.

Participants reported a stronger preference for partners to feel (vs. express) more hope than themselves in stressful situations (M_diff_ = 0.11, 95CI_diff_ = 0.00, 0.22, paired samples *t*(424) = 2.01, *p* = 0.045, *d* = 0.10). Participants did not differ in their preferences for partners to feel vs. express distress (M_diff_ = −0.08, 95CI_diff_ = −0.19, 0.03, paired samples *t*(423) = −1.39, *p* = 0.17, *d* = −0.07), to feel vs. express worry (M_diff_ = 0.05, 95CI_diff_ = −0.04, 0.14, paired samples *t*(424) = 1.16, *p* = 0.25, *d* = 0.06), or to feel vs. express calm (M_diff_ = 0.00, 95CI_diff_ = −0.09, 0.10, paired samples *t*(423) = 0.05, *p* = 0.96, *d* = 0.00).

Participants reported a stronger preference for partners to feel (vs. express) more pride (M_diff_ = 0.11, 95CI_diff_ = 0.03, 0.20, paired samples *t*(424) = 2.58, *p* = 0.010, *d* = 0.13), more hope (M_diff_ = 0.13, 95CI_diff_ = 0.05, 0.20, paired samples *t*(424) = 3.28, *p* = 0.001, *d* = 0.16), and more calm (M_diff_ = 0.16, 95CI_diff_ = 0.08, 0.24, paired samples *t*(424) = 4.05, *p* < 0.001, *d* = 0.16) than themselves in positive situations. Participants did not differ in their preference for partners to feel vs. express excitement (M_diff_ = 0.06, 95CI_diff_ = −0.03, 0.14, paired samples *t*(424) = 1.36, *p* = 0.18, *d* = 0.07).

Aim 1: Preferences for partners’ emotional responses to one’s own stressful situations.

We examined whether participants’ attachment orientation was associated with their preferences for partners’ emotional responses, relative to their own emotional responses, during the participants’ stressful situations (Table 3).

Consistent with Hypothesis 1a, participants with higher attachment anxiety preferred their partners to feel and express more distress, to feel and express more worry, and to express less calm relative to themselves during the participants’ stressful situations. Contrary to Hypothesis 1a, attachment anxiety was not associated with preferences for partners to feel calm, feel hope, or express hope during stressful situations.

Consistent with Hypothesis 1b, participants with higher attachment avoidance preferred their partners to feel and express less hope relative to themselves during the participants’ stressful situations. Contrary to Hypothesis 1b, attachment avoidance was not associated with preferences for partners to feel or express distress, worry, or calm during stressful situations.

Next, we tested the interaction of attachment anxiety × attachment avoidance on preferences for partners’ emotional responses during the participants’ stressful situations. We observed interactions of attachment anxiety × attachment avoidance predicting preferences for partners to feel distress and express worry during the participants’ stressful situations. Examination of marginal slopes for attachment anxiety at low and high levels of attachment avoidance in Figure 1 indicated that: (A) attachment anxiety was more strongly associated with the preference for partners to feel more distress (relative to themselves) during stressful situations when participants also had higher levels of attachment avoidance (fearful avoidance) (*b* = 0.79, *SE* = 0.12, *t* = 6.82, *p* < 0.001, 95CI = 0.56, 1.01) than when participants had lower levels of attachment avoidance (preoccupied attachment) (*b* = 0.32, *SE* = 0.12, *t* = 2.71, *p* = 0.007, 95CI = 0.09, 0.55); and (B) attachment anxiety was associated with the preference for partners to express more worry (relative to themselves) during stressful situations when participants also had higher levels of attachment avoidance (fearful avoidance) (*b* = 0.48, *SE* = 0.11, *t* = 4.29, *p* < 0.001, 95CI = 0.26, 0.71) but not when participants had lower levels of attachment avoidance (preoccupied attachment) (*b* = 0.14, *SE* = 0.11, *t* = 1.26, *p* = 0.21, 95CI = −0.08, 0.37).

There was no interaction of participants’ attachment anxiety × attachment avoidance predicting preferences for partners’ other felt or expressed emotions.

Aim 2: Preferences for partners’ emotional responses to one’s own positive situations.

We examined whether participants’ attachment orientation was associated with their preferences for partners’ emotional responses, relative to their own emotional responses, during the participants’ positive situations (Table 4).

Consistent with Hypothesis 2a, participants with higher attachment anxiety preferred their partners to feel and express more excitement, to feel and express more pride, and to feel more hope than themselves during positive situations. Contrary to Hypothesis 2a, attachment anxiety was not associated with preferences for partners to feel calm, express calm, or express hope during positive situations.

Consistent with Hypothesis 2b, participants with higher attachment avoidance preferred their partners to feel and express less excitement, less pride, and less hope than themselves during the participants’ positive situations. Contrary to Hypothesis 2b, attachment avoidance was not associated with preferences for partners to feel calm or express calm during positive situations.

Next, we tested the interaction of attachment anxiety × attachment avoidance on preferences for partners’ emotional responses during the participants’ positive situations. We observed interactions of attachment anxiety × attachment avoidance predicting preferences for partners to feel and express excitement, to feel and express calm, and to feel pride. Examination of marginal slopes for attachment anxiety at low and high levels of attachment avoidance in Figure 2 indicated that each of these interactions followed a similar pattern.

When participants had high attachment avoidance, higher attachment anxiety was associated with the preference for partners to feel more excitement (*b* = 0.42, *SE* = 0.09, *p* < 0.001, *95CI* = 0.25, 0.60), express more excitement (*b* = 0.33, *SE* = 0.10, *p* = 0.001, *95CI* = 0.14, 0.52), feel more calm (*b* = 0.33, *SE* = 0.10, *p* = 0.001, *95CI* = 0.14, 0.52), express more calm (*b* = 0.22, *SE* = 0.09, *p* = 0.014, *95CI* = 0.05, 0.40), and feel more pride than themselves during the participants’ positive situations (*b* = 0.33, *SE* = 0.10, *p* = 0.001, *95CI* = 0.14, 0.52).

When participants had low attachment avoidance, higher attachment anxiety was not associated with the preference for partners to feel excitement (*b* = 0.12, *SE* = 0.09, *p* = 0.20, *95CI* = −0.06, 0.30), express excitement (*b* = 0.05, *SE* = 0.10, *p* = 0.63, *95CI* = −0.14, 0.24), feel calm (*b* = −0.09, *SE* = 0.10, *p* = 0.34, *95CI* = −0.28, 0.10), express calm (*b* = −0.13, *SE* = 0.09, *p* = 0.17, *95CI* = −30, 0.05), or feel pride (relative to themselves) during the participants’ positive situations (*b* = 0.13, *SE* = 0.09, *p* = 0.15, *95CI* = −0.05, 0.30).

There was no interaction of participants’ attachment anxiety × attachment avoidance predicting preferences for partners to express pride, feel hope, or express hope.

## 6. Discussion

To our knowledge, this study was the first to investigate preferences for partners’ emotional responses to one’s own stressful and positive situations relative to one’s own emotional responses in these situations. This design facilitated interpretation of these preferences in terms of whether they reflected a desire for partners’ emotional responses to amplify, dampen, or maintain individuals’ own emotions in each context. Overall, we found that participants preferred for their partners to dampen their negative emotions in stressful contexts and to amplify their positive emotions in both stressful and positive contexts. We found evidence for most hypothesized associations of attachment orientation and preferences for partners’ emotional responses in each context. Compared to securely attached individuals, anxiously attached individuals preferred their partners to feel and express more negative emotions in stressful situations and more positive emotions in positive situations, while avoidantly attached individuals preferred their partners to feel and express less positive emotions in both stressful situations and positive situations. We also found that individuals with high anxiety and high avoidance (i.e., fearful avoidance) preferred their partners to feel and express more negative emotions in stressful situations and fewer positive emotions in positive situations, whereas individuals with high avoidance and low anxiety (i.e., avoidant dismissing) preferred their partners to feel and express less positive emotions in positive situations. These findings provide a greater understanding of the emotion regulation strategies that individuals prefer to use in their close relationships and how these preferences are shaped by our general attachment working models.

### 6.1. Preferences about Partners’ Emotional Responses to Stressful Situations

Overall, participants preferred their partners to feel and express less distress, less worry, more calm, and more hope than themselves during stressful situations. This suggests that individuals generally wanted their partners to serve as a safe haven that helped them to dampen their negative emotions and amplify their positive emotions in stressful contexts.

Hypothesis 1a was supported. Anxiously attached individuals preferred their partners to feel and express more distress, more worry, and less hope during their own stressful situations. These observations are consistent with attachment theory’s description of anxiously attached individuals as strongly desiring their partner’s validation and approval. Because anxiously attached individuals tend to experience hyperactivation of their own negative emotions in stressful contexts [2], it is possible that these preferences reflect a desire for partners to respond in a way that validates their hyperactivated negative affect and confirms their negative mental representations of themselves and others [25].

Hypothesis 1b was partially supported. Avoidantly attached individuals preferred for their partners to feel and express less hope during their own stressful situations. These preferences are consistent with earlier studies suggesting that attachment avoidance is associated with lower optimism and more catastrophic appraisals of potential threats [15]. Contrary to our initial hypothesis, attachment avoidance was not associated with preferences for partners to feel or express distress or worry during stressful situations. This was surprising given attachment theory’s description of avoidantly attached individuals as being motivated to deactivate their emotions and attachment needs during stressful situations [2]. One possible reason that this hypothesis was not supported is because we observed an interaction of attachment avoidance and attachment anxiety indicating that it was the combination of high attachment avoidance and high attachment anxiety (i.e., fearful avoidance) that was associated with a stronger preference for partners to feel more distress and express more worry during one’s own stressful situations. This observation is consistent with theoretical descriptions of individuals with a fearful-avoidant attachment orientation as engaging in “incoherent” coping strategies that tend to reflect their extreme dysregulation during stressful situations [26]. It is also consistent with theoretical descriptions of fearful-avoidant individuals as being worried about potential signs of rejection from close others [27].

### 6.2. Preferences about Partners’ Emotional Responses to Positive Situations

Overall, participants preferred their partners to feel and express more excitement, pride, and hope than themselves during positive situations. This suggests that individuals generally wanted their partners to serve as a catalyst or secure base for exploration, encouraging their embracing of opportunities and challenges [28]. These preferences are also consistent with the broad literature demonstrating the benefits of interpersonal capitalization in romantic relationships [29].

Hypotheses 2a and 2b were supported. During their own positive situations, anxiously attached individuals preferred their partners to feel and express more excitement, pride, and hope, while avoidantly attached individuals preferred their partners to feel and express less excitement, pride, and hope. These preferences are consistent with earlier theoretical descriptions of anxiously attached individuals as desiring their partner’s validation and approval and of avoidantly attached individuals as wanting to deactivate their emotions and attachment needs [3,27]. They are also consistent with earlier research showing that partners’ capitalization attempts elicit less favorable responses from avoidantly attached individuals and ambivalent responses (i.e., involving feelings of both appreciation and indebtedness) from anxiously attached individuals [29]. Findings for avoidantly attached individuals support earlier work suggesting that attachment avoidance may interfere with the experience of pride in response to a partner’s positive events. Specifically, Mikulincer and Shaver [30] posited that avoidantly attached individuals may experience hubris or hostile envy in response to a partner’s positive events instead of the normative experience of pride.

We also observed interactions of attachment anxiety and attachment avoidance. The combination of high anxiety and high avoidance (i.e., fearful avoidance) was associated with preferences for partners to feel and express more calm in positive situations, while the combination of low anxiety and high avoidance (i.e., avoidant dismissing) was associated with preferences for partners to feel less pride and to feel and express less excitement in positive situations. These observations are consistent with theoretical descriptions of fearful-avoidant individuals and avoidant-dismissing individuals as engaging in coping strategies that are focused on deactivating their emotions and their attachment needs. In theory, these preferences may serve the respective interpersonal and emotion regulation goals of avoidant-dismissing individuals, who are primarily motivated to maintain their autonomy and independence, and fearful-avoidant individuals, who desire closeness with their partners but also fear their potential rejection [27].

### 6.3. Broader Theoretical Contributions

This study provides insight into the types of emotional support that are most desired by different individuals. Our results suggest several novel hypotheses about the effectiveness of different emotional support strategies in stressful and positive situations and how this may differ by general attachment orientation. First, they suggest that individuals will feel most supported in stressful situations when partners dampen their negative emotions; however, this strategy will be less effective for individuals with higher attachment anxiety. Second, they suggest that individuals will feel most supported in positive situations when partners amplify their positive emotions; this strategy will be more effective for individuals with higher attachment anxiety but less effective for individuals with higher attachment avoidance.

Our findings are consistent with Collins and Allard’s [25] assertion that a core function of our attachment working models is to guide our emotional response patterns and how we think about ourselves and others. These findings also raise additional theoretical questions about whether individuals will feel more supported when there is a match between their stated preferences and their partners’ actual emotional responses to stressful and positive situations. Indeed, it is possible that individuals’ preferences accurately reflect the emotional responses that would help them to feel most supported in each context. However, it is equally plausible that these preferences are not necessarily adaptive and instead a reflection of an individual’s general working models of the self and others that developed based on their earlier experiences in close relationships—regardless of whether these models are accurate or adaptive in their current relationship. Consistent with the latter possibility, we found that the emotional responses that anxiously and avoidantly attached individuals preferred from their partners tended to mirror the ways that these individuals respond themselves in stressful and positive situations. For example, anxiously attached individuals, who tend to experience hyperactivation of their own negative emotions during their stressful situations, also preferred for partners to feel and express more distress/worry in these situations. Future observational and experimental studies are needed to disentangle these possibilities by testing whether the individual or dyad benefits when there is a match between individuals’ preferred and received partner emotional responses in each context. Insecure individuals, in particular, may benefit from partners who assist them in regulating their emotions by dampening negative emotions in stressful contexts and by amplifying positive emotions in positive contexts.

Finally, our questionnaire asked participants to differentiate between their preferences for their partners to feel and express emotions. Although the experience and expression of emotions are distinct from a theoretical perspective [31], we did not observe a strong empirical distinction between preferences for partners to feel and express emotions in our study. We measured these preferences separately to account for the possibility that some participants might prefer for their partners to regulate their emotional expressions (i.e., to outwardly exaggerate or downplay their inner feelings) or that dimensions of attachment insecurity would have different associations with preferences for felt and expressed emotions. However, we observed strong correlations between preferences for partners to feel and express specific emotions in each context and found similar associations between attachment orientation and preferences for partners to feel and express specific emotions in each context. One possible reason why we did not observe a stronger distinction between felt and expressed emotions is because our self-report assessment approach was suboptimal for distinguishing between these preferences. As suggested by the strong correlations of felt and expressed emotions and their similar associations with attachment orientation, it is possible that our participants had trouble distinguishing between preferences for their partners’ outward emotional expressions and internal emotional experiences. Future observational studies that assess felt and expressed emotions in both relationship partners could help to distinguish between the impact of partners’ emotional expressions and emotional experiences in these support contexts. Studies assessing relationship partners’ physiological reactivity and physiological synchrony during specific support interactions may also be helpful for addressing these questions [32]. Studies that assess partners’ emotional expressions using observer ratings could also help to distinguish between the impact of felt and expressed emotions on perceived supportiveness and relationship outcomes.

### 6.4. Strengths, Limitations, and Conclusions

Strengths of our study include its investigation of a novel theoretical question based on the predictions of attachment theory in a relatively large and diverse sample of adults in the United States. However, there are also several limitations of this work that represent opportunities for future investigation. First, participants were asked to report how they preferred their partners to respond emotionally in stressful and positive situations. Although self-report methods are important for assessing preferences, the extent to which participants are willing or able to accurately report these preferences is unknown and should be investigated. For example, a future study could aim to replicate these associations by experimentally manipulating how partners respond to disclosures about stressful and positive events in a laboratory discussion. Future research in this area could also examine whether manipulating felt attachment security, such as by using attachment security priming, impacts preferences for partners’ felt or expressed emotions in these contexts. Other future work could comprehensively assess felt and expressed emotions and preferences regarding own and partner emotions from both the disclosing and responding partners, as well as observer ratings of both partners’ emotional expressions, during real-time interactions in the laboratory. Second, our measures asked participants how they preferred their partners to feel and express emotions relative to their own feelings and expressions of emotions. However, it is likely that attachment orientation also influences one’s own emotional responses to stressful and positive events [21]. For example, people with higher attachment avoidance may express lower amounts of hope during positive situations and also prefer that their partners express even less hope than that. Future studies could disentangle this by measuring preferences for partners’ emotions on an absolute measurement scale that is not relative to one’s own emotional feelings and expressions (e.g., assessing preferences for partners’ expressed hope using anchors of not at all and extremely). Third, we prompted participants to reflect on times when they have discussed stressful and positive events with their romantic partner. However, it is unknown whether the types or intensity of events recalled by participants was impacted by their attachment orientation. For example, future studies are needed to test the alternative possibility that anxiously attached individuals preferred their partners to feel more distress because they recalled more intense stressful events than securely attached individuals [10,21]. Future studies could also address this limitation by including measures to assess the type and intensity of events recalled by each participant and statistically controlling for these characteristics.

Overall, these findings contribute to research on attachment theory and interpersonal emotion regulation by demonstrating that attachment insecurity shapes how individuals prefer to regulate their emotions in close relationships in two theoretically important contexts. In general, we observed that anxiously attached individuals’ preferences were consistent with a desire for their partners’ validation and approval, whereas avoidantly attached individuals’ preferences were consistent with a desire to deactivate their emotions and attachment needs. Future research is needed to establish the types of partner emotional responses that will most benefit individuals during their stressful and positive life contexts and over time. Moreover, there is a general need for continued research on preferences for how partners listen and respond to different types of self-disclosures in close relationships [33].

### 6.5. Constraints on Generality

The main boundary condition of this research is its examination of adults from the United States who mostly identified as White or Asian. It is unknown whether similar associations of attachment orientation and preferences for partners’ emotional responses would be observed among adults of other nationalities, cultures, or racial/ethnic backgrounds. Because an earlier review found significant evidence for variation in social support processes (e.g., perceptions of various behaviors as supportive) between individualist and collectivist cultures [34], it will be especially important for future research to evaluate the generalizability of our findings to individuals from more collectivist cultures than the United States. A second potential boundary condition of this research is that our data were collected online in 2020 during the COVID-19 pandemic. Given the impacts of the COVID-19 pandemic on mental health and close relationships, the historical specificity of our observed effects is also unknown. A final potential boundary condition of this research is that our outcome was assessed using newly developed prompts and questionnaires. Certain aspects of these prompts and questionnaires may influence the generalizability of our findings. For example, it is possible that we might have observed more varied associations between attachment orientation and preferences for partners to feel vs. express emotions if this distinction had been made clearer in our prompt. In addition, we assessed individuals’ preferences for their partners to feel and express theoretically relevant positive and negative emotions in each context; it is unknown whether the observed associations would generalize to other types of positive and negative emotions that were not assessed in this study.

## Figures and Tables

**Figure 1 behavsci-14-00077-f001:**
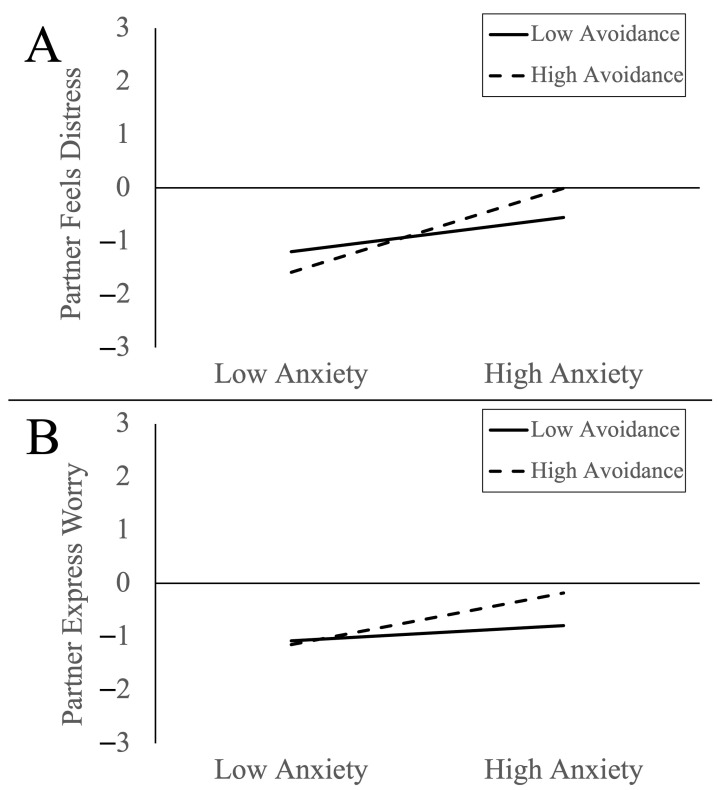
Interaction plots showing preferences for partners to feel distress (**A**) and express worry (**B**) during stressful situations by attachment anxiety and attachment avoidance.

**Figure 2 behavsci-14-00077-f002:**
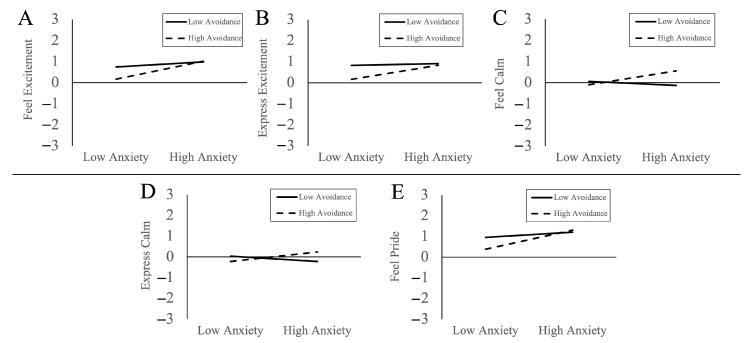
Interaction plots showing preferences for partners to feel excitement (**A**), express excitement (**B**), feel calm (**C**), express calm (**D**), and feel pride (**E**) during positive situations by attachment anxiety and attachment avoidance.

**Table 1 behavsci-14-00077-t001:** Descriptive statistics for study variables.

	M (SD)
Attachment anxiety	3.3 (1.5)
Attachment avoidance	3.6 (1.4)
During stressful situations:	
I want my partner to feel:	
Distress	−0.7 (1.6)
Worry	−0.7 (1.5)
Calm	1.0 (1.4)
Hope	1.3 (1.2)
I want my partner to express:	
Distress	−0.6 (1.6)
Worry	−0.7 (1.5)
Calm	1.0 (1.5)
Hope	1.2 (1.4)
During positive situations:	
I want my partner to feel:	
Excitement	0.8 (1.2)
Pride	1.1 (1.2)
Hope	1.0 (1.2)
Calm	0.2 (1.3)
I want my partner to express:	
Excitement	0.8 (1.3)
Pride	0.9 (1.3)
Hope	0.9 (1.3)
Calm	0.1 (1.2)

**Table 2 behavsci-14-00077-t002:** Bivariate correlations of study variables.

	1	2	3	4	5	6	7	8	9	10	11	12	13	14	15	16	17	18
1. Attachment anxiety	--																	
2. Attachment avoidance	**0.53**	--																
3. Partner feels distress	**0.34**	**0.18**	--															
4. Partner feels worry	**0.28**	**0.18**	**0.55**	--														
5. Partner feels calm	−0.07	−0.08	**−0.43**	**−0.28**	--													
6. Partner feels hope	**−0.11**	**−0.18**	**−0.11**	**−0.19**	**0.32**	--												
7. Partner expresses distress	**0.23**	0.08	**0.74**	**0.52**	**−0.43**	**−0.12**	--											
8. Partner expresses worry	**0.24**	**0.18**	**0.54**	**0.82**	**−0.29**	**−0.17**	**0.56**	--										
9. Partner expresses calm	**−0.16**	**−0.10**	**−0.47**	**−0.30**	**0.76**	**0.35**	**−0.44**	**−0.34**	--									
10. Partner expresses hope	−0.06	**−0.15**	**−0.15**	**−0.16**	**0.30**	**0.62**	**−0.10**	**−0.13**	**0.39**	--								
11. Partner feels excitement	**0.15**	−0.02	**0.43**	**0.30**	**−0.27**	**0.14**	**0.38**	**0.32**	**−0.27**	0.00	--							
12. Partner feels pride	**0.18**	0.01	**0.29**	**0.21**	**−0.10**	**0.22**	**0.29**	**0.26**	**−0.10**	**0.14**	**0.65**	--						
13. Partner feels hope	**0.10**	**−0.12**	**0.28**	**0.23**	**−0.13**	**0.19**	**0.32**	**0.23**	**−0.10**	**0.14**	**0.57**	**0.71**	--					
14. Partner feels calm	**0.13**	**0.12**	**0.12**	**0.15**	**0.15**	−0.01	0.05	**0.11**	0.05	**−0.10**	**0.11**	**0.11**	**0.15**	--				
15. Partner expresses excitement	0.06	−0.09	**0.30**	**0.20**	**−0.18**	**0.12**	**0.34**	**0.23**	**−0.17**	0.06	**0.76**	**0.56**	**0.57**	0.04	--			
16. Partner expresses pride	0.00	**−0.14**	**0.12**	**0.13**	−0.04	**0.19**	**0.23**	**0.14**	0.03	**0.22**	**0.45**	**0.74**	**0.60**	−0.04	**0.55**	--		
17. Partner expresses hope	−0.05	**−0.18**	**0.16**	**0.13**	**−0.12**	**0.17**	**0.28**	**0.17**	−0.03	**0.21**	**0.49**	**0.62**	**0.79**	0.05	**0.51**	**0.63**	--	
18. Partner expresses calm	0.05	0.04	0.04	0.05	**0.12**	−0.04	0.01	0.07	**0.10**	−0.08	0.07	0.07	**0.15**	**0.79**	0.04	−0.06	**0.13**	--

*Note.* Bolded coefficients denote statistical significance at *p* < 0.05.

**Table 3 behavsci-14-00077-t003:** Main effect and interaction of attachment anxiety and avoidance on preferences for partners’ felt and expressed emotions when providing support in stressful situations.

	B	SE	t	*p*	95CI
Partner Feels Distress					
Step 1					
Intercept	−0.72	0.08	−9.62	<0.001	−0.86, −0.57
Attachment anxiety	0.55	0.09	6.30	<0.001	0.38, 0.73
Attachment avoidance	−0.01	0.09	−0.07	0.95	−0.18, 0.17
Step 2					
Attachment anxiety × attachment avoidance	0.24	0.08	3.07	0.002	0.09, 0.39
Partner Feels Worry					
Step 1					
Intercept	−0.66	0.07	−9.28	<0.001	−0.80, −0.52
Attachment anxiety	0.39	0.08	4.59	<0.001	0.22, 0.55
Attachment avoidance	0.07	0.08	0.83	0.41	−0.10, 0.24
Step 2					
Attachment anxiety × attachment avoidance	0.14	0.07	1.91	0.06	−0.00, 0.29
Partner Feels Calm					
Step 1					
Intercept	1.02	0.07	14.92	<0.001	0.88, 1.15
Attachment anxiety	−0.06	0.08	−0.80	0.43	−0.22, 0.09
Attachment avoidance	−0.08	0.08	−0.93	0.35	−0.23, 0.08
Step 2					
Attachment anxiety × attachment avoidance	0.07	0.07	1.02	0.31	−0.07, 0.21
Partner Feels Hope					
Step 1					
Intercept	1.30	0.06	22.56	<0.001	1.19, 1.42
Attachment anxiety	−0.02	0.07	−0.36	0.72	−0.16, 0.11
Attachment avoidance	−0.20	0.07	−2.90	0.004	−0.33, −0.06
Step 2					
Attachment anxiety × attachment avoidance	0.11	0.06	1.88	0.06	−0.01, 0.23
Partner Expresses Distress					
Step 1					
Intercept	−0.64	0.08	−8.53	<0.001	−0.79, −0.49
Attachment anxiety	0.41	0.09	4.69	<0.001	0.24, 0.59
Attachment avoidance	−0.09	0.09	−1.06	0.29	−0.27, 0.08
Step 2					
Attachment anxiety × attachment avoidance	0.02	0.08	0.24	0.81	−0.13, 0.17
Partner Expresses Worry					
Step 1					
Intercept	−0.71	0.07	−9.81	<0.001	−0.85, −0.57
Attachment anxiety	0.32	0.09	3.69	<0.001	0.15, 0.48
Attachment avoidance	0.11	0.09	1.22	0.22	−0.06, 0.27
Step 2					
Attachment anxiety × attachment avoidance	0.17	0.08	2.27	0.02	0.02, 0.32
Partner Expresses Calm					
Step 1					
Intercept	1.01	0.07	14.16	<0.001	0.87, 1.15
Attachment anxiety	−0.21	0.08	−2.52	0.012	−0.38, −0.05
Attachment avoidance	−0.03	0.08	−0.41	0.68	−0.20, 0.13
Step 2					
Attachment anxiety × attachment avoidance	0.08	0.07	1.04	0.30	−0.07, 0.22
Partner Expresses Hope					
Step 1					
Intercept	1.19	0.07	17.51	<0.001	1.06, 1.32
Attachment anxiety	0.03	0.08	0.33	0.74	−0.13, 0.19
Attachment avoidance	−0.22	0.08	−2.76	0.006	−0.38, −0.06
Step 2					
Attachment anxiety × attachment avoidance	0.00	0.07	0.06	0.96	−0.13, 0.14

**Table 4 behavsci-14-00077-t004:** Main effect and interaction of attachment anxiety and avoidance on preferences for partners’ felt and expressed emotions (relative to one’s own emotions) when providing support in positive situations.

	B	SE	t	*p*	95CI
Partner Feels Excitement					
Step 1					
Intercept	0.81	0.06	13.99	<0.001	0.70, 0.93
Attachment anxiety	0.27	0.07	3.97	<0.001	0.14, 0.41
Attachment avoidance	−0.17	0.07	−2.43	0.016	−0.30, −0.03
Step 2					
Attachment anxiety × attachment avoidance	0.15	0.06	2.55	0.011	0.04, 0.27
Partner Feels Proud					
Step 1					
Intercept	1.05	0.06	18.40	<0.001	0.94, 1.16
Attachment anxiety	0.30	0.07	4.41	<0.001	0.17, 0.43
Attachment avoidance	−0.15	0.07	−2.25	0.025	−0.28, −0.02
Step 2					
Attachment anxiety × attachment avoidance	0.17	0.06	2.86	0.004	0.05, 0.28
Partner Feels Hope					
Step 1					
Intercept	0.98	0.06	17.32	<0.001	0.87, 1.09
Attachment anxiety	0.28	0.07	4.13	<0.001	0.14, 0.41
Attachment avoidance	−0.29	0.07	−4.35	<0.001	−0.42, −0.16
Step 2					
Attachment anxiety × attachment avoidance	0.09	0.06	1.53	0.13	−0.03, 0.20
Partner Feels Calm					
Step 1					
Intercept	0.21	0.06	3.39	<0.001	0.09, 0.33
Attachment anxiety	0.12	0.07	1.67	0.10	−0.02, 0.27
Attachment avoidance	0.09	0.07	1.26	0.21	−0.05, 0.24
Step 2					
Attachment anxiety × attachment avoidance	0.21	0.06	3.35	<0.001	0.09, 0.34
Partner Expresses Excitement					
Step 1					
Intercept	0.76	0.06	12.34	<0.001	0.64, 0.88
Attachment anxiety	0.19	0.07	2.63	0.009	0.05, 0.33
Attachment avoidance	−0.21	0.07	−2.94	0.004	−0.36, −0.07
Step 2					
Attachment anxiety × attachment avoidance	0.14	0.06	2.25	0.025	0.02, 0.27
Partner Expresses Pride					
Step 1					
Intercept	0.94	0.06	15.28	<0.001	0.82, 1.06
Attachment anxiety	0.15	0.07	2.01	0.046	0.00, 0.29
Attachment avoidance	−0.26	0.07	−3.62	<0.001	−0.41, −0.12
Step 2					
Attachment anxiety × attachment avoidance	0.03	0.06	.41	0.68	−0.10, 0.15
Partner Expresses Hope					
Step 1					
Intercept	0.85	0.06	14.14	<0.001	0.73, 0.97
Attachment anxiety	0.08	0.07	1.13	0.26	−0.06, 0.22
Attachment avoidance	−0.27	0.07	−3.78	<0.001	−0.41, −0.13
Step 2					
Attachment anxiety × attachment avoidance	0.09	0.06	1.50	0.13	−0.03, 0.22
Partner Expresses Calm					
Step 1					
Intercept	0.05	0.06	0.85	0.40	−0.07, 0.16
Attachment anxiety	0.05	0.07	0.73	0.47	−0.09, 0.19
Attachment avoidance	0.02	0.07	0.23	0.82	−0.12, 0.15
Step 2					
Attachment anxiety × attachment avoidance	0.17	0.06	2.91	0.004	0.06, 0.29

## Data Availability

The data and analysis code for this study are available at: https://osf.io/nkmbw/?view_only=bb9aa20a40fc4e26a89866ed7b264efc.

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
