# Peer review of "Attachment Orientation and Preferences for Partners’ Emotional Responses in Stressful and Positive Situations"

_behavsci, 2024, doi:10.3390/bs14010077_

Round 1
Reviewer 1 Report
Comments and Suggestions for Authors
Review of “Attachment orientation and preferences for partners’ emotional responses in stressful and positive situations” by Chin, Kim, Parsons and Feeney
This is a report of an undergraduate senior thesis conducted during the global
pandemic. It likely had to be conducted within an academic year’s time from start to finish (to fulfil thesis requirements) and to be an online self-report study due to the pandemic. That is understandable. Of course, the facts that participants retrospectively had to recall sharing both negative and positive experiences with partners and then to retrospectively, report with what emotions they wished their partners had responded, represents the weakest part of this work. That is, the researchers and we, as readers, cannot be sure that these reports correspond to actual preferences for partner emotions and expressions in the moment of participants sharing their own positive and negative experiences and feelings.
And yet, I am in favor a accepting this paper for Behavioral Sciences:
1. First, the paper is straightforwardly and clearly written. The introduction is excellent in its review of past relevant literature.
2. The hypotheses are straightforwardly and, I believe, presented despite the study not being pre-registered. The integrity of the senior author (Feeney) is well-known.
3. Most importantly, the topic is novel and important and the results (even if they just represent stated preferences in the moment of the self-reports) are interesting and may well reflect preferences in the moments of sharing stressful and happy events.
That is, these authors address the question of how people who are experiencing and, sometimes, expressing emotions prefer that their partners respond, emotionally. I have not seen prior work on this topic. Relationship researchers really do need to understand preferences for how close partners listen and respond to self-disclosures. (In this regard the authors might be interested in a recent Current Opinion issue edited by Harry Reis and Guy Itzchakov listening and responsiveness.) It is interesting that, in general, people, when stressed, desire partner-listeners to be (and express being) less distressed and more hopeful than they, themselves are when experiencing good things, and desire partner-listeners to be (and to express) more positive emotion than, they, themselves feel and express when they disclose positive things. The individual differences associated with attachment styles are also interesting.
I do not have many suggestions for improving the paper but here are few ideas which the authors might wish to take into account.
1. They might emphasize the limitations of self-report studies just a tad more. I do think that’s the biggest limitation of this work but as long as it’s made clear I also think the study has value. They might suggest new work that would address these issues at the end. Other limitations might still be mentioned but shortened a bit.
2. I found the overarching (collapsing across attachment styles) results to be most interesting. Those overarching results might be highlighted in the abstract where they currently are not mentioned and then a bit more in the paper itself. (Instead, the paper focuses almost exclusively on the attachment findings.)
3. The methodology required people to report how they wanted partners to react in terms of felt and expressed emotion relative to how they, themselves, felt and expressed emotion associated with negative and positive events. Notably, attachment styles probably influenced those baseline feelings (as past literature certainly suggests). I’d suggest addressing this and it means for purposes of interpreting/understanding the results. For instance, for people characterized by anxious attachment, is there a double heightening of negative emotion (they feel more negativity in response to stressful events and they want partners not just to reflect this but to experience and express even more negativity?).
4. Can we be confident that the types and intensity of negative and positive events recalled by participants were equivalent across those with differing attachment orientations? If they did differ, what are the implications for understanding the results of this work?
5. Distinguishing between desired felt emotions in partners and desired expressions of emotion is intriguing. Not too much can be said about the importance of making this distinction given the results and the authors do discuss this well. Yet can more be said about why the authors measured these separately and why, going forward, it makes sense to continue to make this distinction?
Reviewer 2 Report
Comments and Suggestions for Authors
The present manuscript examines partner preferences regarding feelings and expressions of different emotions in response to positive and distressing events in relation to attachment styles.
I have enjoyed reading the manuscript and my impression is that the findings contribute to the knowledge in the field. The analyses and conclusions are sound and I recommend accepting the manuscript pending minor revisions. Congratulations to the authors on their interesting study!
(1) When discussing the retrieval of memories in relation to anxious and avoidant attachment, please consider also adding one or two sentences on studies showing the relation to false memories (see Hudson's works).
Hudson, N. W., & Chopik, W. J. (2023). Seeing you reminds me of things that never happened: Attachment anxiety predicts false memories when people can see the communicator. Journal of Personality and Social Psychology, 124(2), 396–412. https://doi.org/10.1037/pspp0000447
Hudson, N. W., & Fraley, R. C. (2018). Does attachment anxiety promote the encoding of false memories? An investigation of the processes linking adult attachment to memory errors. Journal of Personality and Social Psychology, 115(4), 688–715. https://doi.org/10.1037/pspp0000215
(2) Consider deleting the "Range" column from Table 1 and in-text that all the full range of response options was used.
(3) I suggest to consider the role of gender in the main analyses, as partner preferences are often shown to differ between men and women (e.g., in the field of humor research, women appreciate men that make them laugh whereas men appreciate women who laugh about their humor attempts etc.; Brauer & Proyer, 2019).
Brauer, K., & Proyer, R. T. (2019). Sex differences in attractiveness of humor. Encyclopedia of evolutionary psychological science. Springer. https://doi. org/10.1007/978-3-319-16999-6_3245-1.
Reviewer 3 Report
Comments and Suggestions for Authors
We lose page numbers somewhere along the way so I am referring to the second hypothesis. Specifically, the interaction effects. The first set of results discuss fearful avoidance (high avoidance, high anxiety) and repot significant results for five outcomes (feelings of excitement, pride, and calm; expressions of excitement and calm). The second set presents results for preoccupied avoidance (low avoidance, high anxiety) and reports non-significant results. In the discussion, you mention only that fearful avoidance was associated with feelings and expressions of calm, and that dismissive avoidance (high avoidance, low anxiety) was associated with feelings and expressions of less excitement. Clearly, something here is not lining up. The panels in Figure 2 seem more in line with what is said in the discussion: feelings and expressions of calm look highest for fearful avoidance, feelings and expressions of excitement seem lowest for dismissive avoidance. Pride isn’t really discussed.
